# TRANSFER LEARNING ON KINYARWANDA TWEETS SENTIMENT ANALYSIS

**Roger Byakunda**
African Center of Excellence in Data Science
University of Rwanda
Gikondo, KK 737 Street, Kigali-Rwanda
`221001765@stud.ur.ac.rw`

## ABSTRACT

Pretrained models available on platforms such as Hugging Face have become a valuable resource for the machine learning community, particularly for natural language processing tasks. In this study, we evaluated the performance of Kinyarwanda and English pretrained models for sentiment analysis of Kinyarwanda tweets through transfer learning using Hugging Face pretrained models and Trainer for implementation. We have found that the fine tuned English pretrained models for translated Kinyarwanda tweets dataset using Google translate outperformed Kinyarwanda fine tuned pretrained models.

## 1  INTRODUCTION

Kinyarwanda is a unique and beautiful language spoken in Rwanda, with a rich cultural heritage and a growing presence in the digital world. Despite its significance, Kinyarwanda has been largely overlooked in the field of natural language processing (NLP) due to a lack of available data ready for machine learning tasks(see Orife et al. (2020) for more information). In an effort to address this gap, we conducted a comparative analysis of open source state-of-the-art pretrained models on Kinyarwanda labeled tweets dataset, which are both available on Hugging Face. Our findings shed light on the potential of NLP for Kinyarwanda and provide valuable insights for researchers and developers interested in this exciting and underrepresented language.

## 2  RELATED WORK

Sentiment analysis on African languages, including Kinyarwanda, is a growing area of research, although the number of studies is much lower compared to highly represented languages such as English. Furthermore, many existing studies on African languages are unpublished or published under closed access (see Mesthrie (1995) as cited in Orife et al. (2020) for more information). Muhammad et al. (2022) conducted sentiment analysis on four Nigerian languages by collecting, filtering, processing, and labeling the dataset, and then applying transfer learning using fine tuned pretrained models. Kwaik et al. (2020) employed transfer learning on pretrained models for sentiment analysis on Arabic tweets dataset. In a different study, Bataa & Wu (2019) investigated transfer learning for sentiment analysis on Japanese language using the Rakuten product review and Yahoo movie review datasets. These studies demonstrate the potential of transfer learning and pretrained models for sentiment analysis in underrepresented languages.

## 3  METHODOLOGY

Muhammad et al. (2023) collected a dataset of more than 110,000 annotated tweets covering 14 underrepresented African languages, including Kinyarwanda. We used the dataset, merging the train and validation sets into one and preprocessing the text by removing stopwords, URLs, and emojis, and lowercasing both the train and test sets. For transfer learning to the sentiment analysis task, we used Kinyarwanda pretrained models trained for mask task. In addition, we performed transfer learning on English pretrained models for tweet classification using translated dataset to English

using the Google Translate API. For further exploration we also did preprocessing of translated tweets by removing stopwords(except negative stopwords), punctuation and lastly stemmed. We

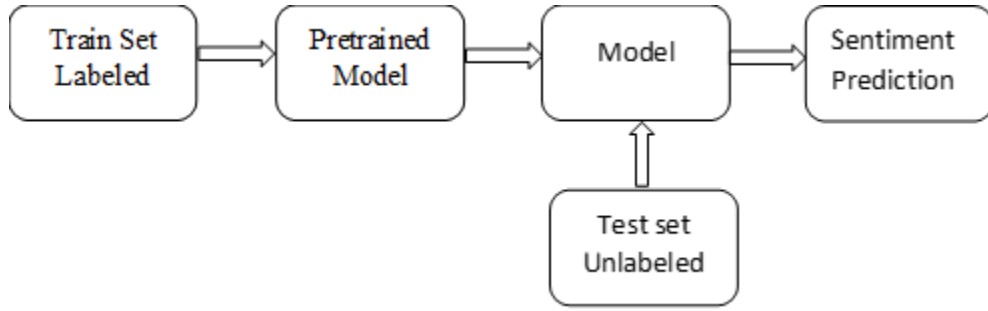

Figure 1: Methodology Flowchart.

evaluated the model's performance by comparing the F1 score of the predicted test set with the actual labeled set, using the trainer framework.

## 4  RESULTS

We conducted fine-tuning of the pretrained models using a standard tutorial as a guide (Amy, 2023). The results show that KinyaBERT-large outperformed xlm-roberta-base-finetuned-kinyarwanda while using Kinyarwanda preprocessed tweets.

| Model | Test Dataset | F1 Score |
|---|---|---|
| bert-base-cased | translated test tweets | 0.661578923 |
| bert-base-cased | preprocessed translated test tweets | 0.625459922 |
| distilbert-base-uncased-finetuned-sst-2-english | translated test tweets | 0.642492736 |
| twitter-xlm-roberta-base-sentiment | translated test tweets | 0.655731011 |
| twitter-roberta-base-sentiment-latest | translated test tweets | **0.686036459** |
| KinyaBERT-large | Preprocessed Kinyarwanda Test Tweets | 0.644029075 |
| xlm-roberta-base-finetuned-kinyarwanda | Preprocessed Kinyarwanda Test Tweets | 0.598704083 |

Table 1: F1 scores of various transformer models on different test datasets.

Additionally, the fine tuned twitter-roberta-base-sentiment-latest model exhibited superior performance compared to other fine tuned pretrained models for the translated dataset. In terms of overall performance, the fine tuned English pretrained models demonstrated better results than Kinyarwanda fine tuned pretrained models for the sentiment analysis task.

## 5  CONCLUSION

Based on the comparative analysis of the pretrained models on Kinyarwanda tweets sentiment analysis through transfer learning task, we can conclude that the fine tuned English pretrained models outperform the Kinyarwanda fine tuned pretrained models. This indicates the importance of having more labeled data and pretrained models in underrepresented African languages like Kinyarwanda. Among the models we experimented with, the fine tuned twitter-roberta-base-sentiment-latest model performed the best with an F1 score of 0.686, closely followed by the fine tuned bert-base-cased model with an F1 score of 0.661 on the test set. However, it is worth noting that the performance of the models could be improved with more fine-tuning and optimization. Overall, our study highlights the need for more research and development of NLP tools and resources for underrepresented African languages, including Kinyarwanda.

## URM STATEMENT

We acknowledge this work meets the URM criteria of ICLR 2023 Tiny Papers Track.

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

## A  APPENDIX

### A.1  ACKNOWLEDGEMENT

We used several pre-trained models for sentiment analysis, including BERT-base-cased, DistilBERT-base-uncased-finetuned-SST-2-English, and Twitter-RoBERTa-base-sentiment-latest, as well as two Kinyarwanda-specific models, KinyaBERT-large and XLM-RoBERTa-base-finetuned-Kinyarwanda see Table 1. **Note**: Pre-trained models and dataset[1] were sourced from the Hugging Face Transformers library (Wolf et al., 2019). Model details and sources: BERT-base-cased (Devlin et al., 2018)[2], DistilBERT-base-uncased-finetuned-SST-2-English (Sanh et al., 2019)[3], Twitter-RoBERTa-base-sentiment-latest[4] and Twitter-XLM-RoBERTa-base-sentiment (Barbieri et al., 2021)[5], KinyaBERT-large (Ishimwe, 2021)[6], XLM-RoBERTa-base-finetuned-Kinyarwanda (Adelani et al., 2021)[7].Lastly we have used Kinyarwanda stopwords (Rubungo, 2020)[8].

---

[1]https://huggingface.co/datasets/shmuhammad/AfriSenti-twitter-sentiment

[2]https://huggingface.co/bert-base-cased

[3]https://huggingface.co/distilbert-base-uncased-finetuned-sst-2-english

[4]https://huggingface.co/cardiffnlp/twitter-roberta-base-sentiment-latest

[5]https://huggingface.co/cardiffnlp/twitter-xlm-roberta-base-sentiment

[6]https://huggingface.co/jean-paul/KinyaBERT-large

[7]https://huggingface.co/Davlan/xlm-roberta-base-finetuned-kinyarwanda

[8]https://github.com/Andrews2017/kkltk

