# OpenReview forum: "Transfer Learning on Kinyarwanda Tweets Sentiment Analysis"
_ICLR.cc/2023/TinyPapers — Submitted to Tiny Papers @ ICLR 2023_

### Official Review · Reviewer_brgq · 2023-04-02

**Confidence:** 5

**Summary Of Contributions:**

The authors compared the performance of a pretrained Kinyarwanda model on Kinyarwanda texts against a pretrained English model on translated kinyarwanda texts. For the English model, the Kinyarwanda texts were translated to English using google translate, the evaluation metric was an F1-score.

**Rating:**

Great Start (GS): a submission which meets some of the reviewing criteria but has room for improvement

**Strengths And Weaknesses:**

**Strengths**
The authors' attempt to compare the performance of a transformer model trained on Kinyarwanda with a larger English-trained model on English translations of Kinyarwanda texts is intriguing.

**Weakness**
As mentioned above, it was intriguing, but the authors failed to properly communicate their results. The result table written by the authors was incomplete and one could not properly compare all the results from their findings.



**Suggested Changes:**

* Proper clarification of their research question and objective with the use of clear and concise language.
* Also provide a thorough review of the related work and related studies.
* A more detailed report of their approach and findings with interpretation and discussion of findings.
* Use proper formatting for your result table.

---

### Official Review · Reviewer_VEnN · 2023-04-03

**Confidence:** 1

**Summary Of Contributions:**

The authors used the Hugging Face Trainer to train a Kinyarwanda tweet classification model by finetuning a range of pre-trained Transformer models. They generated their finetuning dataset using the Google API to translate tweets from Kinyarwanda to English.

**Rating:**

Needs Clarification (NC): a submission which does not meet the reviewing criteria and needs clarification for its described problem or solution

**Strengths And Weaknesses:**

Although an interesting start, the paper appears incomplete, and so it is hard to form any conclusions about the work as a reviewer.

**Suggested Changes:**

Use the most authoritative base-model available, e.g. for ` xlm-roberta-base` use (and cite) the version provided by Facebook, or explain why you have selected a different version of the model.
Work on the formatting of your paper! I would have loved to see/understand the results in the table.

---

### Meta-Review · Area_Chair_BDiN · 2023-04-07

**Recommendation:** Invite to revise
**Confidence:** 5

**Metareview:**

**Summary**
- The paper presents a sentiment classification model for Kinyarwanda by finetuning different pre-trained Transformer models. They used Google Translator to translate tweets from Kinyarwanda to English.

**Strength**
- The paper has an interesting motivation. However, it’s incomplete work.
**Weakness**
- The paper is literally incomplete and doesn’t fulfill any of the CCR requirements.


**Summary:**

 The paper presents a sentiment classification model for Kinyarwanda by finetuning different pre-trained Transformer models. They used Google Translator to translate tweets from Kinyarwanda to English.  Strength: Good motivation. Weakness: Incomplete submission.

**Comments And Feedback To The Authors:**

Please don't submit incomplete work. It would be great to save time for the organizers, reviewers, and everyone.

**Reason For Not Giving A Higher Recommendation:**

The work is incomplete. The authors just submitted a one-page abstract and introduction without any methods or experiments or even a conclusion. So, the work doesn't fulfill any of the CCR requirements.

**Reason For Not Giving A Lower Recommendation:**

N/A

---

### Decision · Program_Chairs · 2023-04-07

Revision accepted; invite to archive